# NoMAD-Attention: Efficient LLM Inference on CPUs Through Multiply-add-free Attention

**Tianyi Zhang**
Dept. of Computer Science,
Rice University
xMAD.ai
Houston, TX
`tz21@rice.edu`

**Jonah Yi**
Dept. of Computer Science,
Rice University
xMAD.ai
Houston, TX
`jwy4@rice.edu`

**Bowen Yao**
Dept. of Computer Science,
Rice University
Houston, TX
`by18@rice.edu`

**Zhaozhuo Xu**
Dept. of Computer Science,
Stevens Institute of Technology
xMAD.ai
Hoboken, NJ
`zxu79@stevens.edu`

**Anshumali Shrivastava**
Dept. of Computer Science, Rice University
Ken Kennedy Institute
ThirdAI Corp.
xMAD.ai
Houston, TX
`anshumali@rice.edu`

## Abstract

Large Language Model (LLM) inference on Central Processing Units (CPU) is challenging due to the vast quantities of Multiply-Add (MAD) matrix operations in the attention computations. This paper highlights a rare gem in modern CPUs, Single-Instruction-Multiple-Data (SIMD) registers, which allows for ultra-low-latency lookups in a batch. We leverage this unique capability to propose NoMAD-Attention, an efficient attention algorithm that replaces MAD operations with in-register lookups. Through hardware-aware algorithmic designs, NoMAD-Attention achieves the computation of attention scores using repeated fast accesses to SIMD registers. NoMAD-Attention works with pre-trained attention-based LLMs without model finetuning. Extensive empirical evaluations demonstrate that NoMAD-Attention maintains the quality of the original LLMs well and speeds up the 4-bit quantized LLaMA-7B-based model by up to $2\times$ at 16k context length.

## 1 Introduction

Auto-regressive transformer-based Large Language Models (LLM) demonstrate remarkable abilities across a wide range of natural language processing tasks without finetuning [35] and exhibit emergent abilities [49] for solving complex problems.

**The Need for Deploying LLM on CPUs.** Despite the potential of LLMs, their deployment is costly [27]. Serving LLMs with billion-scale parameters requires specialized hardware such as NVIDIA A100 Graphics Processing Units (GPUs) [55]. However, mainstream personal devices are predominately equipped with only Central Processing Units (CPUs) [43]. As a result, making LLM-related services accessible to everyone remains a major challenge. Reducing the LLM inference latency on CPUs would significantly influence its accessibility and adoption.

**Expensive Multiply-add Operations for Attention in LLM Inference.** LLM inference on CPUs is compute-bound, and the primary computational bottleneck is the calculation of attention scores [21]. Attention, a mechanism that models token interactions through all-pair dot products, heavily

38th Conference on Neural Information Processing Systems (NeurIPS 2024).

relies on the multiply-add (MAD) kernel on processors. The MAD operation involves computing the product of two numbers and adding that product to an accumulator [44]. MAD plays a crucial role in determining the attention score between tokens and subsequently blending their embeddings within the attention mechanism. The computational cost of attention grows quadratically with the sequence length due to the cumulative MAD operations. Since CPUs have limited parallel cores, they are inefficient for handling highly repetitive and parallel workloads. The extensive MAD operations required by the attention mechanism thus become the primary bottleneck during inference.

**Opportunities and Challenges from Modern CPUs: In-Register Lookups.** The memory hierarchy of modern CPUs has undergone significant evolution, introducing a new type of registers optimized for Single-Instruction-Multiple-Data (SIMD) operations. The SIMD registers vary in size, ranging from 128 bits to 512 bits [41], and support specialized SIMD instructions for high-throughput parallel processing [59]. SIMD registers have become a standard feature in commodity hardware, including laptops and mobile devices [13]. In this context, an in-register lookup (or shuffle) refers to the low-latency random access of information stored within SIMD registers. Storing information such as dot-product lookup tables (LUT) within SIMD registers, as opposed to cache memory, can accelerate LLM inference [4]. However, the limited size of SIMD registers poses challenges to fitting the computational paradigm of existing models.

**Our Proposal: MAD-Free Attention with In-Register Lookups.** Our paper demonstrates a new approach for speeding up LLM inference by leveraging the unique hardware capability of CPUs. We show how the vast quantities of MAD operations in attention computation can be replaced with in-register lookups to mitigate the quadratic computational bottleneck of LLM inference on CPUs. NoMAD-Attention significantly speeds up LLM inference without sacrificing model quality and is compatible with pre-trained attention-based transformers without finetuning.

We summarize our contributions as follows:

1. We identify the extensive MAD operations in attention as the bottleneck of CPU LLM inference and we replace them with fast in-register lookups.
2. We introduce NoMAD-Attention, a MAD-free framework of attention computation for pre-trained attention-based LLMs. NoMAD-Attention leverages hardware-aware algorithmic designs to enable accurate and fast in-register lookup-based estimations of query-key dot products despite the limited capacity of SIMD registers. NoMAD-Attention preserves model quality while yielding considerable speedups over MAD-based attention.
3. Our extensive experiments demonstrate that NoMAD-Attention achieves up to $2\times$ speedup on LLaMA-7B-based models with 4-bit weights at a context length of 16k while maintaining the predictive performance of the original model.

## 2 LLM Inference on CPUs

This section introduces the attention mechanism used in LLMs and the key-value (KV) caching technique for avoiding redundant attention computations. We also discuss the CPU memory hierarchy, which motivates the use of fast in-register lookups.

### 2.1 LLM Attention

Most LLMs are decoder-only attention-based models that are pre-trained on a next token prediction objective. LLMs use masked self-attention, which allows LLMs to cache key and value embeddings to avoid future recomputations. However, this comes at the cost of memory overhead. The autoregressive generation of LLMs consists of two phases: 1. *prompt processing*: the sequence of token embeddings in the prompt is fed through by the model, and their key-value embeddings are cached by the model; and 2. *decoding*: a new token is sampled based on the output embedding of the last token, and the embedding of the new token is fed through the model, the output of which becomes the basis for sampling the next token. The decoding process continues until an end-of-sequence token <EOS> is sampled.

At the decoding step $t$, a single-head masked self-attention computes its output in the following way. The embedding of the current token $e^t$ is transformed into key, query, and value embeddings through distinct transformations $k^t = f_K(e^t), q^t = f_Q(e^t), v^t = f_V(e^t)$.

---

**Algorithm 1** Attention Score Computation in LLM

---
1: **Input:** query $q^t$, key $k^t$, key cache $K_{\text{cache}}^{t-1}$

2: let $K_{\text{cache}}^t \leftarrow \begin{bmatrix} K_{\text{cache}}^{t-1} \\ k^t \end{bmatrix}$                  ▷ Append the current key to key cache

3: return $\text{softmax}\left(\frac{q^t (K_{\text{cache}}^t)^\top}{\sqrt{d}}\right)$

---

Then, the key and value embedding of the current token are appended to the key and value cache, respectively. The KV cache $K_{\text{cache}}^{t-1}, V_{\text{cache}}^{t-1}$ of the step $t-1$ contains the key/value embeddings of all previous tokens, and after appending, the KV cache become

$$K_{\text{cache}}^t = \begin{bmatrix} K_{\text{cache}}^{t-1} \\ k^t \end{bmatrix} = \begin{bmatrix} k^1 \\ k^2 \\ \dots \\ k^t \end{bmatrix}, V_{\text{cache}}^t = \begin{bmatrix} V_{\text{cache}}^{t-1} \\ v^t \end{bmatrix} = \begin{bmatrix} v^1 \\ v^2 \\ \dots \\ v^t \end{bmatrix}$$

Finally, the attention output is computed as

$$\text{attention}(e^t) = \text{softmax}\left( \frac{q^t (K_{\text{cache}}^t)^\top}{\sqrt{d}} \right) V_{\text{cache}}^t$$

where $d$ is the dimensionality of $q^t$. We will refer to the result of $\text{softmax}(\frac{qK^\top}{\sqrt{d}})$ as the attention scores since they dictate how much "attention" each token pays to other tokens. Computations in the prompt processing phase are similar to those in the decoding phase, except all the prompt tokens are computed in batch. LLMs use multi-head attention, which transforms the concatenation of the outputs of multiple single-head attentions to form an output embedding.

**MAD-based Attention.** The attention mechanism models the interaction between tokens by performing all-pair dot products, where each dot product is computed via $d$ Multiply-Add (MAD) operations. Since attention computes the interaction between all pairs of tokens, the amount of MAD operations scales quadratically with the sequence length, quickly overwhelming the computing capability of CPUs. CPUs are designed to handle complex workloads with granular control, while GPUs are optimized for processing simple and repetitive tasks in high throughput. Hence, the success of attention has largely been fueled by the development of highly parallel throughput-oriented processors, such as GPUs [12].

**MAD-based Attention as Bottleneck of LLM Inference.** The computation of attention scores becomes the bottleneck of LLM inference as the sequence length increases (see our analysis in Figure 4). At the $t$-th step of the decoding phase, the time complexity of computing attention score with MAD is $O(t)$ due to $t$ dot products, while all other components of LLMs such as MLP, skip connections, and normalization have a time complexity of $O(1)$. We will focus on optimizing the efficiency of attention score computations in our proposed approach. Algorithm 1 presents the pseudocode for attention score computation, including key caching, for a single-head masked self-attention in LLM. This algorithm will serve as a point of comparison in our proposed approach.

## 2.2 Memory Hierarchy of Modern CPUs

CPU memory is organized into a pyramidal hierarchy, as shown in Figure 1, with faster memory significantly smaller than slower memory. The memory unit with the fastest access speed is the registers. Each compute core accesses its dedicated registers in just 1–2 CPU cycles, but these registers are limited in size, usually not exceeding 64 bits. Modern processors include a new type of registers optimized for Single-Instruction-Multiple-Data (SIMD) operations. These SIMD registers range from 128 bits to 512 bits in size and support a limited set of SIMD instructions for throughput-oriented parallel processing. SIMD registers are common on commodity hardware, including laptops and mobile devices. Using SIMD operations can speed up deep learning models on CPUs by parallelizing matrix multiplications. However, due to the limited number of cores in a CPU, its efficiency in deep learning is still considerably worse than GPU. Prior works [42, 9] have resorted to sparsity and sampling-based approaches, but they require training models from scratch and may not apply to all architectures. Our approach exploits the SIMD registers to shift the computation paradigm from MAD to in-register lookups, which demonstrates significant speedup over MAD-based models.

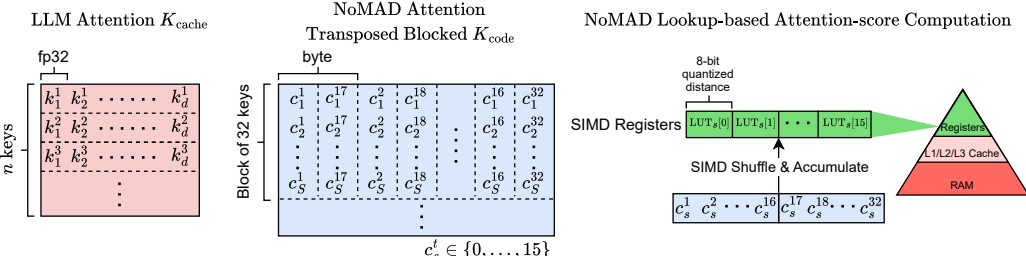

Figure 1: A comparison of memory layouts of the key cache of LLM attention (left) and the key-code cache of NoMAD-Attention (middle), and an illustration of how attention scores are computed through in-register lookups in NoMAD-Attention (right).

When describing our proposed algorithm, we assume SIMD registers are 128 bits wide. Some systems with wider SIMD registers support more parallelism, e.g., 256-bit registers in AVX-2 and 512-bit registers in AVX-512. However, the most universal form of SIMD registers uses 128 bits, which is supported by Arm NEON and AVX-compatible processors.

## 3 Methodology

This section describes our proposed approach, NoMAD-Attention, which replaces MAD operations with in-register lookups to enable faster attention computations on CPUs. NoMAD-Attention utilizes three techniques to enable lookup-based attention: 1. transforming dot product computations to memory lookups through product quantization, 2. compressing lookup tables into SIMD registers for low-latency access, 3. reorganizing the memory layout of key cache for batch parallel dot product lookups.

### 3.1 Transforming Dot-products into Lookups

Previous works have shown that inexact attention scores in transformers work well for sequence modeling [53]. NoMAD leverages Product Quantization (PQ) [23] to compute high-quality estimations of dot products through in-register lookups. PQ, originally designed for compressing high-dimensional vectors, quantizes a floating-point vector into discrete codes. It makes use of *sub-quantizers*; for a $d$-dimensional vector space, a vector is divided evenly in dimension into $S$ sub-vectors, where each sub-vector has dimension $d_{\text{sub}} = \frac{d}{S}$, and each sub-vector space is quantized independently. We use $\pi_s(e)$, where $s \in \{1 \dots S\}$, to denote the function that maps a $d$-dimensional vector $e$ to its $d_{\text{sub}}$-dimensional sub-vector of the $s$-th sub-quantizer. *Codebooks* are used to quantize sub-vectors to codes, which are collections of cluster centroids learned from a set of calibration vectors. We use $b_{s,c}$ to denote the $c$-th centroid in the codebook of the $s$-th sub-quantizer. For a given vector $e$, the product-quantized codes of $e$, denoted $c_1, \dots, c_S$, are the indexes of the nearest centroid of each sub-quantizer, i.e.,

$$\text{PQ}(e) = [c_1 \dots c_S], \text{where } c_s = \arg\min_c \left\| \pi_s(e) - b_{s,c} \right\|$$

Once base vectors have been product-quantized to codes, PQ leverages asymmetric distance computation to keep estimation errors low. In the computed distances, the original query vector is used while the quantized base vectors are used, hence the asymmetry. For a given query $q$, the distances to the centroids of each sub-quantizer are computed and stored in a lookup table (LUT). Then, the corresponding distances in the LUT are looked up based on the codes of base vectors and accumulated to produce the final distance estimation. More concretely, denoting the distance between query $q$ and the $c$-th centroid for the $s$-th sub-quantizer using $\text{LUT}_s[c] = \text{dist}\big(\pi_s(q), b_{s,c}\big)$, then the estimated distance between query $q$ and a product-quantized base vector $e$, where $\text{PQ}(e) = [c_1 \dots c_S]$, is

$$\widetilde{\text{dist}}(q, e) = \sum_{s=1}^{S} \text{LUT}_s[c_s]$$

Building upon previous work [7], we employ product quantization to approximate dot products within the attention mechanism. We product-quantize the key vectors in attention to produce key codes, which will be stored in place of the key cache of LLM attention. The codebooks are learned by performing clustering on a set of key vectors from a calibration set. The key vectors are quantized to the nearest centroid with respect to L2 distance. For a given query, the query-dependent LUT is computed to hold dot products with respect to centroids. Dot products of sub-vectors are retrieved from the LUT based on key codes and accumulated to produce the final dot product estimates. This procedure allows us to compute attention scores through lookups.

## 3.2 Compressing Lookup Tables into SIMD Registers

Estimating dot products through PQ mostly eliminates the use of MAD kernels in the computation of attention scores. However, this approach yields limited speedup over dot-product attention since a high proportion of the CPU cycles are wasted due to cache/memory access stalling. L1-cache-resident LUT is not enough to offer high-performance PQ [3]. The full potential of lookup-based attention can only be unlocked by having the LUT stored in registers, which take only 1-2 CPU cycles to access. However, the highly limited size of registers poses a challenge to fitting the LUT. In PQ, each sub-quantizer commonly uses 256 centroids, which translates to 8-bit codes. Combined with 32-bit floating-point (FP32) dot products, the LUT for each sub-quantizer consumes 8192 bits of memory, while the SIMD registers are only 128 bits wide. We leverage hardware-aware techniques proposed in [4] to enable low-latency retrieval from register-resident LUT.

**8-bit Quantized Dot Products in LUT** Due to the 128-bit width of SIMD registers, the FP32 representation of dot product is too costly to store. Adopting FP32 dot products in LUT implies that each codebook can only contain up to 4 centroids, which will lead to significant quantization errors. Therefore, we adopt the 8-bit dynamically quantized representation of dot products. Compressing beyond 8-bit is infeasible since most SIMD instruction sets do not support parallel lookups below 8 bits. The quantization is done dynamically for each query to minimize quantization errors. For a given query and sub-quantizer, dot products to centroids are first computed in full FP32 precision. Then, the quantization range is determined by the minimum and maximum dot products to the centroids. Finally, the range is evenly divided into $2^8$ buckets, and dot products are quantized to the bucket they fall into. More formally, suppose $\mathrm{dp_{min}} = \min_c(\pi_s(q) \cdot b_{s,c})$ and $\mathrm{dp_{max}} = \max_c(\pi_s(q) \cdot b_{s,c})$ are the minimum and maximum dot products of the query $q$ to the centroids of the $s$-th sub-quantizer, then the LUT stores the quantized dot products to centroid $c$ as

$$\mathrm{LUT}_s[c] = \left\lfloor \frac{(\pi_s(q) \cdot b_{s,c}) - \mathrm{dp_{min}}}{(\mathrm{dp_{max}} - \mathrm{dp_{min}})/(2^8 - 1)} \right\rfloor \tag{1}$$

The quantization and de-quantization process can be done efficiently without much computational overhead, and the quantization error is kept low thanks to dynamic query-dependent quantization (we analyze its effects in Table 3).

**Constrained Codebook Size** By adopting 8-bit quantized dot products in LUT, we can fit 16 dot products on 128-bit SIMD registers. This implies that the codebook size of each sub-quantizer is constrained to 16 centroids, and evidence suggests this limited size may work well with attention: it has been shown that the output of attention loses rank extremely quickly [15], implying that the intermediate embeddings of transformers may exhibit clear clustering structures.

## 3.3 Reorganizing Key Cache Memory Layout

Quantized dot products and constrained codebooks enable LUT to be stored in SIMD registers, but the layout format of the key cache needs to be reorganized for SIMD instructions. The original key cache in LLM attention stores each key vector contiguously in a row to optimize single vector reads. NoMAD-Attention uses the key-code cache in place of the key cache, which stores the quantized codes of keys. To allow fast lookups of LUT entries based on key codes, we store the key codes in a transposed blocked format. A comparison between the LLM key cache and the NoMAD key-code cache is given in Figure 1.

The storage format of the NoMAD key-code cache is *transposed*: stored in column-major order instead of row-major, and *blocked*: with 32 keys as a block. The SIMD instruction shuffle, leveraged for performing low-latency batch lookups, takes a batch of byte-size integers as input and

**Algorithm 2** NoMAD-Attention Score Computation
___
1: **Input:** query $q^t$, key $k^t$, key-code cache $K_{\text{code}}^{t-1}$
2: let $c_s^t \leftarrow \arg\min_{c \in \{0,\dots,15\}} \|\pi_s(k^t), b_{s,c}\|_2$ for $s = 1 \dots S$      ▷ Compute codes for the current key
3: let $K_{\text{code}}^t \leftarrow$ insert $c_s^t$ into $K_{\text{code}}^{t-1}$ for $s = 1 \dots S$
                                 ▷ Insert codes of the current key into the key-code cache
4: let $\text{LUT}_s[c] \leftarrow \text{quantize}(\pi_s(q^t) \cdot b_{s,c})$ for $s = 1 \dots S, c = 0 \dots 15$
                                 ▷ Store 8-bit quantized dot products (Equation 1) in LUT
5: let $\text{accu}[1 \dots t] \leftarrow 0$                                       ▷ Initialize accumulators
6: **for** $i \leftarrow 1 \dots \lceil \frac{t}{32} \rceil$ **do**                       ▷ Perform in-register lookups in batch of 32 keys
7:     **for** $s \leftarrow 1 \dots S$ **do**
8:         $\texttt{simd\_load}(\text{LUT}_s)$                           ▷ Load LUT into registers
9:         $\text{accu}[32i - 31 \dots 32i] \leftarrow$
                $\texttt{simd\_add}\big(\text{accu}[32i - 31 \dots 32i], \texttt{simd\_shuffle}(\text{LUT}_s, K_{\text{code}}^{32i-31\dots32i,s})\big)$
10:     **end for**
11: **end for**
12: return $\text{softmax}\big(\frac{\text{dequantize}(\text{accu}[1\dots t])}{\sqrt{d}}\big)$
___

retrieves the values held in the registers corresponding to the integer indices. The original storage format of the key cache stores all dimensions of a key contiguously, which precludes efficient use of `shuffle`. To maximize the usage of the LUT held in registers, we store key codes belonging to the same sub-quantizer contiguously in rows of 32 codes. Since `shuffle` performs lookups in a batch size of 16, the keys within the same block are stored in alternating order. Each quantized code occupies half a byte as there are 16 centroids in a codebook, while the `shuffle` instruction uses each byte as an input argument. By performing SIMD bit-shifting and bit-masking on a block of alternating keys, we obtain the key codes in the original order, ready for use with `shuffle`. Section A in the Appendix describes additional details on how `shuffle` is performed on each block of key-code cache, and provides pseudocode.

### 3.4 NoMAD-Attention

By combining these three techniques, NoMAD-Attention achieves fast MAD-free attention score computations through SIMD in-register lookups. For a given query, first, LUTs with 8-bit quantized dot products are computed for each sub-quantizer. Then, an LUT is loaded into registers, followed by SIMD `shuffle` instructions to retrieve dot products in the LUT in batch based on key codes. The loading and lookup are repeated for all sub-quantizers, and the retrieved dot products are accumulated in batch through SIMD `add`. Finally, the quantized dot products accumulated over all sub-quantizers are de-quantized, scaled, and fed through softmax to produce the attention scores. The pseudocode for NoMAD-Attention score computations is given in Algorithm 2.

### 3.5 Learning Key Compression

Compressing each segment of key activation embeddings into 4-bit codes, without degrading model quality, is challenging. The straightforward approach is to first cache key activation embeddings on a calibration dataset, and then learn the centroids through k-means clustering [29] on the embeddings. However, this first-cut approach significantly degrades model quality, especially for $d_{\text{sub}} > 1$ (we analyze its effects in Section 4.2). The first-cut approach performs sub-optimally since it is uninformed; clustering is performed with the aim to minimize the reconstruction error with all tokens. However, as shown previously [60, 28], the key cache of certain tokens are more pivotal for preserving model quality. Therefore, we leverage the Fisher Information Matrix (FIM) to bias the centroids towards important activations. Specifically, we approximate the Hessian using diagonals of the FIM, which is the element-wise square of the gradient, and use it to weigh the reconstruction errors. Using the FIM to minimize reconstruction error was first proposed in [26]. Our optimization objective for

Table 1: Perplexity on WikiText-2 and C4 and accuracy on 6 benchmarks of LLMs with Attention and NoMAD-Attention.

| | | WT-2↓ | C4↓ | SciQ↑ | Arc-E↑ | Arc-C↑ | Hellaswag↑ | WinoGrande↑ | PIQA↑ | **Avg.↑** |
|---|---|---|---|---|---|---|---|---|---|---|
| LLaMA-7b | Attention | 5.68 | 7.08 | 94.6 | 75.21 | 41.89 | 56.93 | 70.09 | 78.67 | 69.57 |
| | NoMAD-Attention ($d_{sub} = 1$) | 5.74 | 7.14 | 94.9 | 75.34 | 41.81 | 56.57 | 70.56 | 78.56 | 69.62 |
| | NoMAD-Attention ($d_{sub} = 2$) | 6.11 | 7.56 | 93.3 | 73.65 | 38.65 | 54.45 | 67.56 | 77.86 | 67.58 |
| | NoMAD-Attention ($d_{sub} = 4$) | 9.23 | 12.66 | 84.4 | 66.41 | 32.59 | 46.74 | 59.75 | 74.32 | 60.70 |
| LLaMA-13b | Attention | 5.09 | 6.61 | 95.0 | 77.40 | 46.42 | 59.93 | 72.85 | 79.16 | 71.79 |
| | NoMAD-Attention ($d_{sub} = 1$) | 5.14 | 6.65 | 95.1 | 77.15 | 46.67 | 59.74 | 72.85 | 79.22 | 71.79 |
| | NoMAD-Attention ($d_{sub} = 2$) | 5.44 | 6.96 | 94.8 | 76.43 | 44.37 | 58.05 | 71.82 | 78.18 | 70.61 |
| | NoMAD-Attention ($d_{sub} = 4$) | 8.19 | 10.66 | 89.1 | 73.53 | 37.71 | 51.19 | 60.93 | 76.71 | 64.86 |
| LLaMA-2-7b | Attention | 5.47 | 6.97 | 94.0 | 76.30 | 43.43 | 57.16 | 69.06 | 78.07 | 69.67 |
| | NoMAD-Attention ($d_{sub} = 1$) | 5.53 | 7.02 | 93.9 | 75.80 | 42.92 | 56.71 | 69.30 | 77.97 | 69.43 |
| | NoMAD-Attention ($d_{sub} = 2$) | 5.97 | 7.54 | 93.2 | 73.48 | 39.76 | 54.71 | 67.96 | 76.93 | 67.67 |
| | NoMAD-Attention ($d_{sub} = 4$) | 10.19 | 13.24 | 85.1 | 68.77 | 33.70 | 47.11 | 56.51 | 74.27 | 60.91 |
| LLaMA-2-13b | Attention | 4.88 | 6.47 | 94.6 | 79.42 | 48.46 | 60.07 | 72.30 | 79.00 | 72.31 |
| | NoMAD-Attention ($d_{sub} = 1$) | 4.92 | 6.50 | 94.8 | 79.00 | 47.87 | 59.81 | 71.51 | 78.94 | 71.99 |
| | NoMAD-Attention ($d_{sub} = 2$) | 5.24 | 6.85 | 94.2 | 77.61 | 45.39 | 58.46 | 70.72 | 77.75 | 70.69 |
| | NoMAD-Attention ($d_{sub} = 4$) | 8.22 | 10.86 | 90.8 | 72.81 | 37.80 | 51.24 | 55.96 | 76.33 | 64.16 |

learning the centroids for the $s$-th sub-quantizer is

$$b_{s,0}^{\star} \ldots b_{s,15}^{\star} = \underset{b_{s,0}\ldots b_{s,15}\in\mathbb{R}^{d_{sub}}}{\arg\min} \sum_{c=0}^{15}\sum_{i=1}^{n} w_i^c \left\|\pi_s(k_i) - b_{s,c}\right\|_2^2,$$

$$\text{where } w_i^c = \begin{cases} \overbrace{\text{grad}(\pi_s(k_i))^\top \text{grad}(\pi_s(k_i))}^{\text{partial sum of diagonals of FIM}} & \text{if } c = \arg\min_a \|\pi_s(k_i) - b_{s,a}\|_2 \\ 0 & \text{otherwise} \end{cases} \quad (2)$$

where $k_1 \ldots k_n$ are cached key activation embeddings on the calibration dataset. We use weighted k-means++ [5] to optimize the objective.

# 4    Experiments

This section evaluates the effectiveness of our proposed NoMAD-Attention in preserving model quality and speeding up LLM inference on CPUs. We first introduce the software, hardware, models, benchmarks, and baselines, then provide detailed results and discussions, and finally perform an ablation study to validate each component of our proposal.

**Software and Hardware** Our implementation of NoMAD-Attention is built in C and C++, based on the open-source projects llama.cpp [20] and FAISS [16]. We also built a GPU implementation of NoMAD-Attention for quick prototyping and key-compression learning, which is based on PyTorch [33] and HuggingFace Transformers [51]. Experiments for latency and throughput are performed on a Linux server equipped with an Intel Xeon E5-2695 V3 14-core CPU, which supports AVX2 SIMD instructions, and 512GB of DDR4 RAM. Experiments for accuracy and perplexity are performed on two NVIDIA A100-40GB GPUs.

**Models and Benchmarks** We evaluate the quality of NoMAD-Attention with 4 popular LLMs: 1. LLaMA-7b 2. LLaMA-13b [45] 3. LLaMA-2-7b 4. LLaMA-2-13b [46]. We measure the model quality with perplexity on WikiText-2 [30] and C4 [14] at the context length of 2048, and zero-shot accuracy (using the default configurations of lm-evaluation-harness [19]) on SciQ [50], Arc Easy (Arc-E), Arc Challenge (Arc-C) [11], Hellaswag [54], WinoGrande [38], and PIQA [6]. The centroids for key compression of NoMAD-Attention are learned on a calibration set of 16 sequences from WikiText-2, each with 2048 tokens. To test the model efficiency, we benchmark the latency and throughput of CodeLlama-7b [37] (with 16-bit weights and 4-bit q4_0 quantized weights), which has a longer context length of 16,384 than the LLaMA family of models. We compare the efficiency of NoMAD-Attention-based models (with $d_{sub} \in \{1, 2, 4\}$) against Attention-based models with a llama.cpp-based implementation.

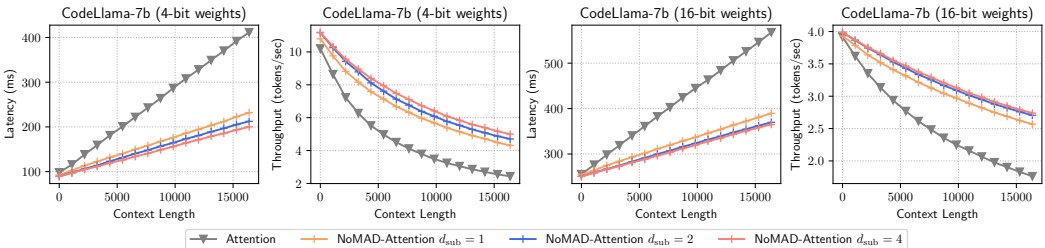

Figure 2: Latency and throughput of decoding for CodeLlama-7b (4-bit and 16-bit weights) with Attention and NoMAD-Attention. NoMAD-Attention achieves 1.78–2.07× higher throughput than Attention with 4-bit CodeLlama-7b, and 1.46–1.56× higher throughput with 16-bit CodeLlama-7b.

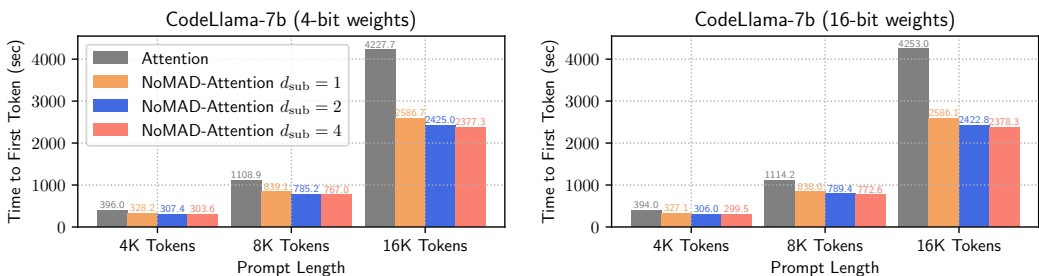

Figure 3: Time for processing prompts of different lengths for CodeLlama-7b (4-bit and 16-bit weights) with Attention and NoMAD-Attention. NoMAD-Attention achieves 1.63–1.79× speedup over Attention to process a prompt of 16k tokens.

## 4.1 Results

**Model Quality** Table 1 presents the perplexity and accuracy of NoMAD-Attention on different benchmarks with comparisons to Attention. NoMAD-Attention ($d_{sub} = 1$) incurs negligible perplexity and accuracy loss, $d_{sub} = 2$ incurs minimal degradation, and $d_{sub} = 4$ incurs some degradation due to the high key compression factor.

**Decoding Efficiency** The latency and throughput for decoding 16,384 tokens for NoMAD-Attention-based and Attention-based CodeLlama-7b are presented in Figure 2. For 4-bit CodeLlama-7b, NoMAD-Attention achieves 1.78×, 1.95×, and 2.07× higher throughput (tokens per second) than Attention for $d_{sub} = 1, 2, 4$, respectively. For 16-bit CodeLlama, NoMAD-Attention achieves 1.46×, 1.54×, and 1.56× higher throughput (tokens per second) than Attention for $d_{sub} = 1, 2, 4$, respectively.

**Prompt Processing Efficiency** We investigate the efficiency of NoMAD-Attention for prompt processing. We generate prompts of length 4,000, 8,000, and 16,000 tokens and record the time until the first token is decoded. Figure 3 presents prompt processing time for CodeLlama-7b. At the prompt length of 16k tokens, NoMAD-Attention speeds up prompt processing time by $1.63 - 1.78×$ for the 4-bit model, and $1.64 - 1.79×$ for the 16-bit model. Prompt processing times are similar for 16-bit and 4-bit model, since weight quantization is ineffective for speeding up batch processing.

## 4.2 Ablation Study

We perform a set of ablative experiments to study the latency breakdown, and the effects of FIM-informed clustering and 8-bit LUT quantization on model quality.

**Latency Breakdown** We investigate the makeup of latency in each decoding step. Figure 4 presents the latency breakdown of 4-bit CodeLlama-7b for decoding 16,384 tokens, and Figure 7 in the appendix shows the latency breakdown for 16-bit CodeLlama-7b. The latency of linear projections

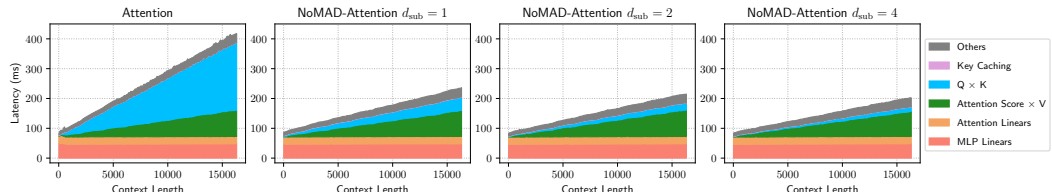

Figure 4: The breakdown of decoding latency of 4-bit CodeLlama-7b for Attention and NoMAD-Attention. NoMAD-Attention effectively reduces the latency of computing query-key dot-products by 5.24–14.77$\times$ over Attention.

in MLP and Attention stays constant as context length increases, and the latency of key caching is insignificant compared to other components. The latency of dot-products between queries and keys, as well as the multiplications between attention scores and values, grows linearly with increasing context length. In the original Attention, computing attention scores quickly becomes the latency bottleneck as context length increases. NoMAD-Attention mitigates this bottleneck by speeding up query-key dot-product computation by 5.24$\times$, 9.75$\times$, and 14.77$\times$ for $d_{\text{sub}} = 1, 2, 4$, respectively.

**FIM-Informed Clustering** We study the effects of FIM-informed centroid learning for preserving model quality. Table 2 presents the perplexity of NoMAD-Attention-based LLaMA-7b on WikiText-2 with uninformed centroids and FIM-informed centroids. FIM-informed centroids consistently achieve better model quality, shown by the lower perplexity.

**LUT Quantization** We examine the effects of 8-bit quantized dot products in LUT on model quality. Table 3 presents the perplexity of NoMAD-Attention-based LLaMA-7b on WikiText-2 with 8-bit quantized LUT and 32-bit unquantized LUT. 8-bit quantization of LUT incurs negligible loss in perplexity.

Table 2: Ablative experiments on the effects of FIM-informed centroid learning on the perplexity of LLaMA-7b on WikiText-2.

|  | Perplexity $\downarrow$ | |
|  | Uninformed | FIM-informed |
| --- | --- | --- |
| NoMAD-Attention ($d_{\text{sub}} = 1$) | 5.76 | 5.74 |
| NoMAD-Attention ($d_{\text{sub}} = 2$) | 7.05 | 6.11 |
| NoMAD-Attention ($d_{\text{sub}} = 4$) | 21.39 | 9.23 |
| Attention | 5.68 | |

Table 3: Ablative experiments on the effects of quantized LUT on the perplexity of LLaMA-7b on WikiText-2.

|  | Perplexity $\downarrow$ | |
|  | 8-bit LUT | 32-bit LUT |
| --- | --- | --- |
| NoMAD-Attention ($d_{\text{sub}} = 1$) | 5.74 | 5.74 |
| NoMAD-Attention ($d_{\text{sub}} = 2$) | 6.11 | 6.10 |
| NoMAD-Attention ($d_{\text{sub}} = 4$) | 9.23 | 9.23 |
| Attention | 5.68 | |

## 5 Related Works

**Approximate Attention and Efficient Transformer** Since the introduction of attention in transformers [47], there has been a body of work on approximating the attention mechanism for efficient training and inference. Dynamically sparse attention was achieved using LSH [24], Nyström method [52], and random sampling [53]. Low-rank attention has also been extensively explored [48, 10, 8] and shown to have compute- and memory-efficiency advantages. Hardware-aware attention mechanisms such as FlashAttention [12] propose to mitigate the IO bottleneck in GPUs. In large language models, multiple approaches [60, 28, 58] have been proposed to reduce the high memory overhead of the KV cache. For CPU-only environments, [39] proposes to speed up LLM inference through weight quantization. Weight quantization [18, 56] is an effective approach for accelerating LLM inference and fine-tuning [57] by mitigating the IO bottleneck.

**Matrix Multiplication Optimization and Compression** Approximate matrix multiplication is applicable in a wide range of computational problems, and its optimization has been a topic of interest for years [32]. Modern researchers have begun optimizing matrix multiplication around the specific limitations of computers, including mitigating the IO bottleneck between the CPU and main memory [25]. Compression techniques were developed to accelerate large-scale matrix multiplications and

scale up the size of multiplication [31, 7, 25, 1]. However, many of these algorithms still had limitations that make them too inaccurate or costly for approximating matrix multiplications in LLMs [7, 25].

# 6 Conclusion

This study addresses the challenges of large language model inference on Central Processing Units (CPUs), particularly the difficulties associated with the expensive Multiply-Add (MAD) matrix operations in attention mechanisms. The investigation highlighted the untapped potential of Single-Instruction-Multiple-Data (SIMD) registers and their fast in-register lookup capabilities within CPUs. The proposed NoMAD-Attention algorithm serves as an efficient alternative to traditional MAD-based approaches, leveraging in-register lookups and optimizing memory access to SIMD registers. The implementation of NoMAD-Attention resulted in a significant acceleration of LLaMA-7B-based model inference, achieving up to a $2\times$ speedup on CPUs.

## Acknowledgements

This work was supported by National Science Foundation SHF-2211815, Ken Kennedy Institute, and grants from Adobe and VMware.

## Limitations and Broader Impacts

Our proposed method is targeted towards CPUs with SIMD capabilities, and may not generalize to other types of processors. We make LLMs more accessible on commodity hardware, which contributes to the democratization of artificial intelligence and reduces the carbon footprints. Other than the negative societal impacts already presented by LLMs, we expect no additional negative impacts from our work.

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

## Appendix / Supplemental Material

## A  Details Regarding SIMD Instructions

The SIMD `shuffle` presented in Algorithm 2 is a simplification of the actual hardware implementation. We give full details of lines 5 to 11 in Algorithm 2 in Algorithm 3. Keys are stored in blocks of 32, in which keys are stored in an alternating order (see Figure 1 for an illustration). After a LUT is loaded into registers, the row of key codes in the block corresponding to the sub-quantizer is used to perform `shuffle`. First, each byte in the row is bit-shifted to the right by 4 bits via a SIMD instruction, which produces the codes of the first 16 keys in the block. The codes are fed to `shuffle` to retrieve the quantized dot products of the first 16 keys from the LUT. Then, the first 4 bits of each byte in the row are masked out via a SIMD instruction, which produces the code of the last 16 keys in the block. They are similarly used to retrieve the quantized dot products from the LUT. The retrieved quantized dot products of 32 keys are accumulated in the accumulator. Since quantized dot products are 8 bits wide, accumulating them in 8-bit accumulators easily results in overflows. Therefore, 16-bit accumulators are used to accumulate quantized dot products.

---
**Algorithm 3** NoMAD Dot-Product Lookup Accumulation Loop
---

1: let $\text{accu}[1\ldots t] \leftarrow 0$                                       ▷ Initialize 16-bit unsigned accumulators
2: **for** $i \leftarrow 1\ldots\lceil\frac{t}{32}\rceil$ **do**
3:    **for** $s \leftarrow 1\ldots S$ **do**
4:       $\text{simd\_load}(\text{LUT}_s)$                                  ▷ Load LUT into registers
5:       let $K_{\text{cache}}^{32i-31\ldots32i-16,s} \leftarrow \text{simd\_bitwise\_right\_shift}(K_{\text{cache}}^{32i-31\ldots32i,s}, 4)$   ▷ Obtain the first 15 key codes through bit shifting
6:       $\text{accu}[32i-31\ldots32i-16] \leftarrow \text{simd\_add}\big($
          $\text{accu}[32i-31\ldots32i-16],$
          $\text{simd\_shuffle}(\text{LUT}_s, K_{\text{cache}}^{32i-31\ldots32i-16,s})\big)$
7:       let $K_{\text{cache}}^{32i-15\ldots32i,s} \leftarrow \text{simd\_bitwise\_and}(K_{\text{cache}}^{32i-15\ldots32i,s}, \text{0xf})$   ▷ Obtain the last 15 key codes through bitwise and
8:       $\text{accu}[32i-15\ldots32i] \leftarrow \text{simd\_add}\big($
          $\text{accu}[32i-15\ldots32i],$
          $\text{simd\_shuffle}(\text{LUT}_s, K_{\text{cache}}^{32i-15\ldots32i,s})\big)$
9:    **end for**
10: **end for**

---

## B  Visual Explanations on $K_{\text{cache}}$ and Lookup Table (LUT) Construction

Figure 5 illustrates the process of mapping and compressing key vector $k^t$ to construct $K_{\text{cache}}^t$. For an input key vector $k^t$, functions $\pi_s$, where $s \in 1\ldots S$, split the vector into sub-vectors $k^t = (\pi_1(k^t), \pi_2(k^t), \ldots, \pi_S(k^t))$. Subsequently, each sub-quantizer $\pi_s(k^t)$ is mapped to its nearest centroid $c_s^t$ by referencing the codebook $b_s$, where $i \in 1\ldots S$, among 16 centroids in the codebook. The resulting values are then stored in the key cache $K_{\text{cache}}^t$.

Similarly, Figure 6 illustrates the process of mapping and compressing query vector $q^t$ to construct the Look-up Tables (LUT). Given a query vector $q^t$, functions $\pi_s$, where $s \in 1\ldots S$, first split the query into sub-queries $q = (\pi_1(q^t), \pi_2(q^t), \ldots, \pi_S(q^t))$. Subsequently, the distances between each sub-query $\pi_s(q^t)$ and the 16 centroids from the codebook $b_s$ are computed and then quantized to values within the range of 0–255. Lastly, the quantized vectors are converted into 8-bit codes and stored in $\text{LUT}_s^t$.

## C  Overhead of Centroid Learning and Storage

Table 4 details the time overhead for learning centroids, which involves saving activations and gradients, and weighted k-means on the saved embeddings. NoMAD-Attention has low learning overheads and can easily scale to larger models.

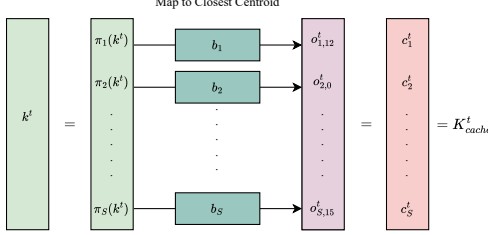

Figure 5: Illustration demonstrating the mapping of an input key $k^t$ to its $s$-th sub-quantizer using $\pi_s(k^t)$, where $s \in 1 \ldots S$. Subsequently, each sub-quantizer maps to its closest centroid $c_i^t$, where $i \in 1 \ldots S$, and the results are stored in the key cache $K_{\text{cache}}^t$.

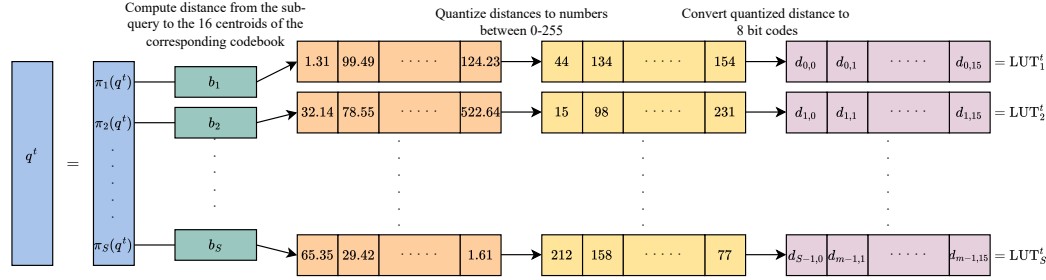

Figure 6: Illustration depicting the mapping of a query vector $q$ to its $s$-th sub-quantizer using $\pi_s(q)$, where $s \in 1 \ldots S$. Subsequently, the distance between $\pi_s(q)$ and 16 centroids is computed. This distance is quantized to a value within the range of 0-255, and the resulting quantized distance is further converted into 8-bit codes, which are stored in $LUT_s$.

The learned centroids of NoMAD-Attention introduce a small amount of memory overhead. The storage overhead of centroids can be calculated as $l \times h \times d \times 16 \times 4$ bytes, where $l$ represents the number of layers, $h$ the number of attention key heads, $d$ the dimensionality of each attention head, 16 the number of centroids, and 4 the number of bytes for storing each centroid parameter. Therefore, the codebook memory overhead for LLaMA-7b/LLaMA-2-7b with $d_{\text{sub}} = \{1, 2, 4\}$ is 8.4MB, and the memory overhead for LLaMA-13b/LLaMA-2-13b with $d_{\text{sub}} = \{1, 2, 4\}$ is 13.1MB, which is a modest memory footprint compared to the overall model size.

Table 4: Overhead of learning centroids for key compression.

| Model | NoMAD Config. | Saving Activations & Gradients | Weighted K-means |
|---|---|---|---|
| LLaMA-7b | $d_{\text{sub}} = 1$ | 4 mins | 27 mins |
| | $d_{\text{sub}} = 2$ | 4 mins | 14 mins |
| | $d_{\text{sub}} = 4$ | 4 mins | 7 mins |
| LLaMA-13b | $d_{\text{sub}} = 1$ | 8 mins | 42 mins |
| | $d_{\text{sub}} = 2$ | 8 mins | 22 mins |
| | $d_{\text{sub}} = 4$ | 8 mins | 11 mins |
| LLaMA-2-7b | $d_{\text{sub}} = 1$ | 4 mins | 27 mins |
| | $d_{\text{sub}} = 2$ | 4 mins | 14 mins |
| | $d_{\text{sub}} = 4$ | 4 mins | 7 mins |
| LLaMA-2-13b | $d_{\text{sub}} = 1$ | 8 mins | 42 mins |
| | $d_{\text{sub}} = 2$ | 8 mins | 23 mins |
| | $d_{\text{sub}} = 4$ | 8 mins | 11 mins |

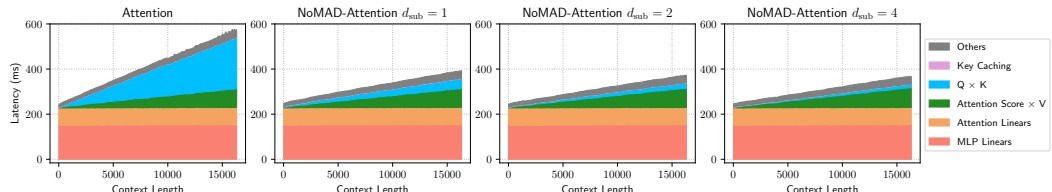

Figure 7: The breakdown of decoding latency of 16-bit CodeLlama-7b for Attention and NoMAD-Attention. NoMAD-Attention effectively reduces the latency for computing dot-products of queries and keys.

## D  Latency Breakdown

Figure 7 presents the latency breakdown of CodeLlama-7b with 16-bit weights.

## E  Comparison with Integer Quantization

We empirically compare NoMAD-Attention and Attention with integer quantized key cache. We test the accuracy of LLaMA-7b, using NoMAD-Attention ($d_{\text{sub}} = 1$) and 8-bit and 4-bit integer quantized key cache (q8_0 and q4_0 in llama.cpp). Furthermore, we test the decoding latency at a context length of 16K tokens, and measure the speedup compared to the full model. As shown in Table 5, NoMAD-Attention ($d_{sub} = 1$) demonstrates better model quality than INT4-quantized key cache and significantly higher speedup than INT4- and INT8-quantized key cache.

Table 5: Zero-shot accuracy and decoding latency (at a context length of 16K) comparison between NoMAD-Attention and Attention with integer quantized key cache, for LLaMA-7b.

|  | SciQ | Arc-E | Arc-C | Hellaswag | WinoGrande | PIQA | Avg | Latency Per Token (ms) | Speedup |
|---|---|---|---|---|---|---|---|---|---|
| Original Attention | 94.6 | 75.21 | 41.89 | 56.93 | 70.09 | 78.67 | 69.565 | 572.497 | - |
| INT8 Key Cache | 94.7 | 75.29 | 42.15 | 57.00 | 70.09 | 78.63 | 69.643 | 562.377 | 1.018× |
| INT4 Key Cache | 93.6 | 74.33 | 41.04 | 55.34 | 67.96 | 77.69 | 68.327 | 540.319 | 1.060× |
| NoMAD-Attention ($d_{\text{sub}} = 1$) | 94.9 | 75.34 | 41.81 | 56.57 | 70.56 | 78.56 | 69.623 | 391.098 | **1.464×** |

## F  Results on Llama-3

We evaluate the effectiveness of NoMAD-Attention on the Llama-3-8b [17] model, which uses grouped query attention [2]. Table 6 presents the accuracy of Llama-3-8b with Attention and NoMAD-Attention across a range of downstream tasks. The results demonstrate that NoMAD-Attention with $d_{sub} = 1$ effectively maintains model quality for Llama 3.

Table 6: Accuracy of the Llama-3-8b model with NoMAD-Attention on downstream tasks.

|  | SciQ | Arc-E | Arc-C | Hellaswag | WinoGrande | PIQA |
|---|---|---|---|---|---|---|
| Attention | 96.4 | 80.09 | 50.51 | 60.18 | 72.77 | 79.71 |
| NoMAD-Attention ($d_{\text{sub}} = 1$) | 96.1 | 80.05 | 49.49 | 59.86 | 73.16 | 79.65 |
| NoMAD-Attention ($d_{\text{sub}} = 2$) | 94.8 | 78.32 | 46.59 | 57.52 | 70.17 | 78.35 |
| NoMAD-Attention ($d_{\text{sub}} = 4$) | 86.3 | 70.08 | 37.54 | 47.38 | 57.38 | 76.61 |

## G  Additional Evaluations

We evaluate NoMAD-Attention on the more challenging MMLU [22], GPQA [36], and MGSM (English) [40] benchmarks. We use lm-evaluation-harness [19] for accuracy evaluation, and the task names are `mmlu_stem`, `mmlu_social_sciences`, `mmlu_humanities`, `mmlu_other`, `gpqa_main_zeroshot`, `mgsm_direct_en`. As shown in Table 7, NoMAD-Attention with $d_{sub} = 1$ effectively maintains model quality across these diverse challenging tasks.

Table 7: Accuracy of LLaMA models on the challenging MMLU, GPQA, and MGSM (English) benchmarks.

| Model | Method | MMLU | | | | GPQA | MGSM |
|-------|--------|------|----------------|-----------|-------|------|------|
| | | STEM | Social Sciences | Humanities | Other | | |
| LLaMA-7b | Attention | 26.39 | 29.57 | 29.73 | 33.15 | 20.98 | 4.8 |
| | NoMAD-Attention ($d_{sub} = 1$) | 27.31 | 29.12 | 29.44 | 32.8 | 23.66 | 4.0 |
| | NoMAD-Attention ($d_{sub} = 2$) | 25.25 | 24.99 | 26.82 | 30.09 | 20.08 | 2.0 |
| | NoMAD-Attention ($d_{sub} = 4$) | 25.21 | 22.98 | 24.85 | 26.94 | 25.67 | 1.6 |
| LLaMA-13b | Attention | 34.13 | 44.39 | 40.60 | 46.48 | 28.35 | 7.2 |
| | NoMAD-Attention ($d_{sub} = 1$) | 33.43 | 43.71 | 39.57 | 45.83 | 28.35 | 7.6 |
| | NoMAD-Attention ($d_{sub} = 2$) | 30.70 | 37.96 | 34.39 | 40.20 | 27.01 | 6.8 |
| | NoMAD-Attention ($d_{sub} = 4$) | 25.88 | 25.58 | 27.46 | 27.55 | 25.67 | 1.2 |

# H    Clarification on $d_{\text{sub}}$

NoMAD-Attention achieves speedup relative to full Attention when $d_{\text{sub}} = 1$ primarily due to two factors: 1. **Leveraging Lower-Latency, Higher-Throughput Instructions.** Unlike the vanilla multiply-add attention, which relies on batched multiplication and addition SIMD instructions (e.g., `_mm256_mul_ps` and `_mm256_add_ps` in AVX2), NoMAD-Attention utilizes the SIMD lookup instruction (`_mm256_shuffle_epi8`). This latter instruction operates on more elements at once (32 elements versus 8) and exhibits lower latency (1 cycle vs. 4 cycles on most architectures) [1], contributing significantly to the efficiency gains. 2. **Minimized Data Movement.** The product-quantized key cache employed in NoMAD-Attention effectively reduces the volume of data transferred between RAM and registers, hence speeding up computations.

# I    Generalizing to More Attention Types

NoMAD-Attention can generalize to other pretrained transformer models and attention variants such as grouped-query attention (GQA) [2] and Attention with Linear Biases (ALiBi) [34]. Given that GQA employs a shared key head across multiple query heads, we can adapt NoMAD-Attention to reuse the same key codes for performing in-register lookups. The ALiBi method adds a linear bias term to the query-key dot products, a process fully compatible with the NoMAD-Attention approach.

---

[1] `https://www.intel.com/content/www/us/en/docs/intrinsics-guide/index.html`

