# OpenReview forum: "NoMAD-Attention: Efficient LLM Inference on CPUs Through Multiply-add-free Attention"
_NeurIPS.cc/2024/Conference — NeurIPS 2024 poster_

### Official Review · Reviewer_hE7X · 2024-07-14

**Soundness:** 3
**Presentation:** 3
**Contribution:** 3
**Rating:** 6
**Confidence:** 4

**Summary:**

The paper presents NoMAD-Attention, using  SIMD registers in CPU, to speedup LLM inference in CPU.

**Strengths:**

(1) Important problem with interesting solution.
(2) The idea is straightforward, and the figure is very clear.
(3) Good system speedup and maintaining the original performance of attention.

**Weaknesses:**

The ML benchmarks seem a bit weak (e.g. perplexity, and PIQA-like easy benchmark). Can you evaluate on harder tasks like MT-Bench?

**Questions:**

Please see weakness.

**Limitations:**

The authors include the limitation that this only works on CPUs with SIMD, but the reviewer thinks this is not a significant limitation.

---

> ### Author Rebuttal · Authors · 2024-08-07
>
> We greatly appreciate the reviewer's careful review and valuable suggestions. We address their comments as follows.
>
> **[W1] Evaluations on harder tasks.**
>
> To further assess NoMAD-Attention, we conducted additional evaluations on the more challenging MMLU, GPQA, and MGSM (English) benchmarks. Our results demonstrate that NoMAD-Attention with $d_{sub}=1$ effectively maintains model quality across these diverse tasks.
>
> |                                     | MMLU (STEM) | MMLU (Social Sciences) | MMLU (Humanities) | MMLU (Other) | GPQA  | MGSM |
> |-------------------------------------|-------------|------------------------|-------------------|--------------|-------|------|
> | LLaMA-7b (Attention)                | 26.39       | 29.57                  | 29.73             | 33.15        | 20.98 | 4.8  |
> | LLaMA-7b (NoMAD-Attention $d_{sub}=1$)  | 27.31       | 29.12                  | 29.44             | 32.8         | 23.66 | 4.0    |
> | LLaMA-7b (NoMAD-Attention $d_{sub}=2$)  | 25.25       | 24.99                  | 26.82             | 30.09        | 20.08 | 2.0    |
> | LLaMA-7b (NoMAD-Attention $d_{sub}=4$)  | 25.21       | 22.98                  | 24.85             | 26.94        | 25.67 | 1.6  |
> | LLaMA-13b (Attention)               | 34.13       | 44.39                  | 40.60              | 46.48        | 28.35 | 7.2  |
> | LLaMA-13b (NoMAD-Attention $d_{sub}=1$) | 33.43       | 43.71                  | 39.57             | 45.83        | 28.35 | 7.6  |
> | LLaMA-13b (NoMAD-Attention $d_{sub}=2$) | 30.70        | 37.96                  | 34.39             | 40.20         | 27.01 | 6.8  |
> | LLaMA-13b (NoMAD-Attention $d_{sub}=4$) | 25.88       | 25.58                  | 27.46             | 27.55        | 25.67 | 1.2  |

---

> > ### Comment · Reviewer_hE7X · 2024-08-10
> >
> > Thank you! This is good. I raised the score to 6. Please consider accept this paper.

---

### Official Review · Reviewer_z7Yd · 2024-07-14

**Soundness:** 4
**Presentation:** 4
**Contribution:** 4
**Rating:** 7
**Confidence:** 4

**Summary:**

This paper proposes using SIMD instructions on CPUs to speed up Transformers by removing multiply-add instructions. The paper replaces the attention operation a lookup-table based alternative. The paper motivates the application well for Transformer inference on CPUs.

**Strengths:**

Strengths:
* Important problem (efficiency) on an understudied platform (CPUs)
* Shows speedup compared to standard attention
* The quality studies are strong, with evaluation on upstream and downstream benchmarks. There is some degradation in quality, but it can be adjusted based on hyperparameters.

I have also reviewed this paper in a previous conference (ICML) and I see that the paper is much improved from that version. I recommend accept based on the strength of the motivation and quality of the evaluation.

**Weaknesses:**

Evaluation on additional and more recent open models (LLaMa-2, LLaMa-3, Gemma, QWEN) would make the paper more convincing. Two open questions from the current evaluation are how well the method generalizes to other pretrained models, and whether changes in the attention algorithm from more recent models change it.

These are not necessary for the paper, but are natural follow-ups for the work and would improve the paper's impact.

**Questions:**

How well does the method generalize to other open models and to changes in the attention algorithm?

**Limitations:**

Yes

---

> ### Author Rebuttal · Authors · 2024-08-07
>
> We sincerely thank the reviewer for their support of our paper and the thoughtful suggestions. We address the reviewer's concerns as follows.
>
> **[W1] Evaluation on additional and more recent open models.**
>
> - We conducted additional experiments on the LLaMA-3-8b model with NoMAD-Attention across a range of downstream tasks. Our results demonstrate that NoMAD-Attention with $d_{sub}=1$ effectively maintains model quality.
>
> |                                      | SciQ | Arc-E | Arc-C | Hellaswag | WinoGrande | PIQA  |
> |--------------------------------------|------|-------|-------|-----------|------------|-------|
> | LLaMA-3-8b (Attention)               | 96.4 | 80.09 | 50.51 | 60.18     | 72.77      | 79.71 |
> | LLaMA-3-8b (NoMAD-Attention d_sub=1) | 96.1 | 80.05 | 49.49 | 59.86     | 73.16      | 79.65 |
> | LLaMA-3-8b (NoMAD-Attention d_sub=2) | 94.8 | 78.32 | 46.59 | 57.52     | 70.17      | 78.35 |
> | LLaMA-3-8b (NoMAD-Attention d_sub=4) | 86.3 | 70.08 | 37.54 | 47.38     | 57.38      | 76.61 |
>
> **[W2, Q1] How well does the method generalize to other open models and to changes in the attention algorithm?**
>
> NoMAD-Attention can generalize to other pretrained models and attention variants such as grouped-query attention (GQA) [1] and Attention with Linear Biases (ALiBi) [2]. Given that GQA employs shared key heads across multiple query heads, we can adapt NoMAD-Attention to reuse the same key codes for performing in-register lookups. The ALiBi method adds a linear bias term to the query-key dot products, a process fully compatible with the NoMAD-Attention approach.
>
> **References**
>
> [1] Ainslie, Joshua, et al. "GQA: Training Generalized Multi-Query Transformer Models from Multi-Head Checkpoints." Proceedings of the 2023 Conference on Empirical Methods in Natural Language Processing. 2023.
>
> [2] Press, Ofir, Noah Smith, and Mike Lewis. "Train Short, Test Long: Attention with Linear Biases Enables Input Length Extrapolation." International Conference on Learning Representations.

---

> > ### Comment · Reviewer_z7Yd · 2024-08-13
> > **Response**
> >
> > Thank you for the rebuttal - I will be keeping my high score. Good work!

---

### Official Review · Reviewer_CZ4j · 2024-07-14

**Soundness:** 3
**Presentation:** 3
**Contribution:** 2
**Rating:** 6
**Confidence:** 5

**Summary:**

This paper utilizes product quantization (PQ) to replace dot product operations in the matrix multiplications involved in the attention mechanism of transformers with memory lookup operations, showcased on language models. To my knowledge, this technique has been first introduced by Blalok et al (reference [5] in the paper). This paper goes further into the systems aspect of deploying PQ on GPUs by specifically utilizing SIMD registers to store the codebook used in PQ. This promises to significantly improve the memory access overhead even compared to alternative implementations that keep the PQ codebook in the L1 cache. However, this poses very stringent codebook size constraints that are addressed in the paper. PQ is done completely post-training unlike prior work, but uses a calibration dataset.

**Strengths:**

- This paper is addressing an important topic: LLM optimization on commodity CPUs.
- The use of SIMD registers is innovative and results in significant speedup.
- PQ has been demonstrated a number of times in the past couple of years, but never with strong results on LLMs.

**Weaknesses:**

- d_sub is very confusing to me. What is the length or your codeword? if d_sub = 1, is the size of the codeword = 1? how is that product quantization? How does it result in a speedup?
- the proposed product quantization has been proposed to replace MAD in  matrix multiplications in reference [5]. Line 148 in this paper is overclaiming its contribution suggesting that this submitted paper has proposed this method.
- Can you please clarify the additional memory overhead introduced by the codebook?
- "double quantization" of PQ has been introduced in prior work and is not cited: Abouelhamayed et al: PQA: Exploring the Potential of Product Quantization in DNN Hardware Acceleration.

**Questions:**

see above

---

> ### Author Rebuttal · Authors · 2024-08-07
>
> We are grateful for the reviewers' careful review and insightful comments. We have addressed their feedback in detail below.
>
> **[W1] d_sub is very confusing to me. What is the length or your codeword? if d_sub = 1, is the size of the codeword = 1? how is that product quantization? How does it result in a speedup?**
>
> - We are sorry for the confusion. To clarify, $d_{sub}$ represents the dimensionality of each sub-quantizer, which is equivalent to the dimensionality of each cluster centroid or the number of dimensions encoded by each product-quantized key code. Notably, the special case where $d_{sub}=1$ can be considered a generalization of multi-dimensional product quantization.
> - $d_{sub}=1$ results in a speedup primarily due to two reasons: **1. Leveraging Lower-Latency, Higher-Throughput Instructions.** Unlike the vanilla multiply-add attention, which relies on batched multiplication and addition SIMD instructions (e.g., `_mm256_mul_ps` and `_mm256_add_ps` in AVX2), NoMAD-Attention utilizes the SIMD lookup instruction (`_mm256_shuffle_epi8`). This latter instruction operates on more elements at once (32 elements versus 8) and exhibits lower latency (1 cycle vs. 4 cycles on most architectures) [1], contributing significantly to the efficiency gains. **2. Minimized Data Movement.** The product-quantized key cache employed in NoMAD-Attention effectively reduces the volume of data transferred between RAM and registers, hence speeding up computations.
>
> **[W2] Line 148 in this paper is overclaiming its contribution.**
>
> - We apologize for this oversight. We will revise line 148 in the final paper to the following: "Building upon previous work [5], we employ product quantization to approximate dot products within the attention mechanism."
>
> **[W3] Can you please clarify the additional memory overhead introduced by the codebook? --- Sure!**
>
> - The memory overhead introduced by the codebook can be computed as $l \times h \times d \times 16 \times 4$ bytes, where $l$ represents the number of layers, $h$ the number of attention key heads, $d$ the dimensionality in an attention head, $16$ the number of centroids, and $4$ the number of bytes for storing each centroid parameter.
> - Therefore, the codebook memory overhead for LLaMA-7b/LLaMA-2-7b with $d_{sub} \in \\{1,2,4\\}$ is 8.4MB, and the memory overhead for LLaMA-13b/LLaMA-2-13b with $d_{sub} \in \\{1,2,4\\}$ is 13.1MB, which is a relatively modest memory footprint compared to the overall model size.
> - We thank the reviewer for highlighting this point, and we will incorporate a detailed discussion of codebook memory overhead into the final paper.
>
> **[W4] "double quantization" of PQ has been introduced in prior work and is not cited.**
>
> - We thank the reviewer for bringing PQA to our attention. We will incorporate a citation to this relevant work in the final version of our paper.
>
> **References**
>
> [1] https://www.intel.com/content/www/us/en/docs/intrinsics-guide/index.html

---

> > ### Comment · Reviewer_CZ4j · 2024-08-07
> > **Thank you for the response**
> >
> > The clarifications proposed by the authors should improve the paper writeup.
> > I will maintain my positive-leaning score for this paper..

---

### Official Review · Reviewer_QJwg · 2024-07-15

**Soundness:** 3
**Presentation:** 2
**Contribution:** 2
**Rating:** 6
**Confidence:** 3

**Summary:**

This paper proposes an algorithm to compute vector inner product efficiently on CPU by exploiting the fast access speed of in-register memory for fast self-attention computation on CPU for model inference. Instead of use multiply and add to compute the inner product between query and key vectors, the authors propose to break down the inner product between two vectors to the summation of multiple inner product between sub-vectors. For each sub-vector product, given a query vector, the algorithm enumerates all possible dot-product results with all possible key sub-vectors in a lookup table during preprocessing. The key sub-vectors are quantized so that the the lookup table is small enough to fit into the in-register memory. The authors further optimize the efficiency via optimizing key cache memory layout.

**Strengths:**

1. The algorithm uses lookup operation to avoid multiplication and allows calculating multiple dimensions at once when the dimension of sub-vectors is larger than 1.
2. The algorithm is able to utilize an extremely small memory (128 bits) to accelerate the calculation.

**Weaknesses:**

1. It would be better to also give some data about the latency of GPU decoding so that we know how far CPU is behind the GPU in Figure 2.
2. When the $d_{sub} > 1$, as shown in Table 1, the performance of these models got a severe hit, so it seems that the algorithm only works well on enumerating all possible scalar products for $d_{sub} = 1$.
3. Even when $d_{sub} = 1$, there are still latency improvement on Figure 2, but I think the number of operations should be the same as the vanilla multiply-add attention, except that each multiply is replaced with a lookup. I am wondering where is the efficiency improvement.
4. For $d_{sub} = 1$, the algorithm is quite relevant to quantized matmul for self-attention computation, I am wondering how the method performs compared to int8 or lower bit width quantization in terms of model accuracy and efficiency.

**Questions:**

see weakness section.

**Limitations:**

The authors discussed limitations and societal impact.

---

> ### Author Rebuttal · Authors · 2024-08-07
>
> We sincerely appreciate the reviewers' careful consideration of our paper and their valuable feedback. We have addressed each of their comments below.
>
> **[W1] It would be better to also give some data about the latency of GPU decoding so that we know how far CPU is behind the GPU in Figure 2.**
>
> - We thank the reviewer for their valuable suggestion. We will incorporate the latency of GPU decoding into the final version of the paper to provide a more comprehensive evaluation.
>
> **[W2 & W3] Values of $d_{sub}=1$.**
> - As demonstrated in Table 1, a $d_{sub}$ value of 2 still yields reasonable model performance, as measured by perplexity and accuracy.
> - The $d_{sub}=1$ configuration already achieves significant speedups compared to the original model, with a 1.78x improvement observed at a context length of 16K (Figure 2).
> - The efficiency improvement achieved when $d_{sub}=1$ is primarily due to two factors: **1. Leveraging Lower-Latency, Higher-Throughput Instructions.** Unlike the vanilla multiply-add attention, which relies on batched multiplication and addition SIMD instructions (e.g., `_mm256_mul_ps` and `_mm256_add_ps` in AVX2), NoMAD-Attention utilizes the SIMD lookup instruction (`_mm256_shuffle_epi8`). This latter instruction operates on more elements at once (32 elements versus 8) and exhibits lower latency (1 cycle vs. 4 cycles on most architectures) [1], contributing significantly to the efficiency gains. **2. Minimized Data Movement.** The product-quantized key cache employed in NoMAD-Attention effectively reduces the volume of data transferred between RAM and registers, hence speeding up computations.
>
> **[W4] How does the method perform compared to int8 or lower bit width quantization in terms of model accuracy and efficiency? --- Here is a comparison.**
>
> - We present additional experiments comparing NoMAD-Attention with INT8 and INT4 key cache quantization (q8_0 and q4_0 in llama.cpp). We report the accuracy on various benchmarks and the decoding latency at a context length of 16K tokens for LLaMA-7b.
> - NoMAD-Attention ($d_{sub}$=1) demonstrates better model quality than INT4 quantized key cache and significantly higher speedup than INT4 and INT8 quantized key cache.
>
> |                        | SciQ | Arc-E | Arc-C | Hellaswag | WinoGrande | PIQA  | Avg    | Decoding Latency (16K) | Speedup |
> |------------------------|------|-------|-------|-----------|------------|-------|--------|---------------------------------------|---------|
> | Original Attention     | 94.6 | 75.21 | 41.89 | 56.93     | 70.09      | 78.67 | 69.565 | 572.497                               | -       |
> | INT8 Quantized Key Cache | 94.7 | 75.29 | 42.15 | 57.00        | 70.09      | 78.63 | 69.643 | 562.377                               | 1.018x   |
> | INT4 Quantized Key Cache | 93.6 | 74.33 | 41.04 | 55.34     | 67.96      | 77.69 | 68.327 | 540.319                               | 1.060x   |
> | NoMAD (d_sub=1)        | 94.9 | 75.34 | 41.81 | 56.57     | 70.56      | 78.56 | 69.623 | 391.098                               | **1.464x**   |
>
> **References**
>
> [1] https://www.intel.com/content/www/us/en/docs/intrinsics-guide/index.html

---

### Author Rebuttal · Authors · 2024-08-07

We sincerely appreciate the reviewers' careful evaluation of our paper and their valuable feedback. In the following section, we address common concerns raised by multiple reviewers. We are happy to provide further clarification during the discussion period.

**1. Regarding $d_{sub}$**

NoMAD-Attention achieves speedup when $d_{sub}=1$ primarily due to two factors: **1. Leveraging Lower-Latency, Higher-Throughput Instructions.** Unlike the vanilla multiply-add attention, which relies on batched multiplication and addition SIMD instructions (e.g., `_mm256_mul_ps` and `_mm256_add_ps` in AVX2), NoMAD-Attention utilizes the SIMD lookup instruction (`_mm256_shuffle_epi8`). This latter instruction operates on more elements at once (32 elements versus 8) and exhibits lower latency (1 cycle vs. 4 cycles on most architectures) [1], contributing significantly to the efficiency gains. **2. Minimized Data Movement.** The product-quantized key cache employed in NoMAD-Attention effectively reduces the volume of data transferred between RAM and registers, hence speeding up computations.

**References**

[1] https://www.intel.com/content/www/us/en/docs/intrinsics-guide/index.html

---

### Decision · Program_Chairs · 2024-09-25

**Decision:**

Accept (poster)

**Comment:**

This paper presents an effective method for accelerating LLMS on commodity CPUs, and the reviews clearly indicate acceptance.